# The Current Strategy in Hormonal and Non-Hormonal Therapies in Menopause—A Comprehensive Review

**DOI:** 10.3390/life13030649

**Published:** 2023-02-26

**Authors:** Anca Lucia Pop, Bogdana Adriana Nasui, Roxana Georgiana Bors, Ovidiu Nicolae Penes, Ana Gabriela Prada, Eliza Clotea, Simona Crisan, Calin Cobelschi, Claudia Mehedintu, Monica Mihaela Carstoiu, Valentin Nicolae Varlas

**Affiliations:** 1Faculty of Pharmacy, “Carol Davila” University of Medicine and Pharmacy, 020945 Bucharest, Romania; 2Department of Community Health, “Iuliu Hațieganu” University of Medicine and Pharmacy, 400349 Cluj-Napoca, Romania; 3Department of Obstetrics and Gynecology, Filantropia Clinical Hospital, 011132 Bucharest, Romania; roxana_georgiana20@yahoo.com (R.G.B.); valentin.varlas@umfcd.ro (V.N.V.); 4Department of Intensive Care, “Carol Davila” University of Medicine and Pharmacy, University Clinical Hospital, 050474 Bucharest, Romania; 5Department of Geriatrics and Gerontology, “St. Lukae’s” Hospital Chronic Diseases, 041915 Bucharest, Romania; 6R&D Center, AC HELCOR, Babes St. No 50, 430092 Baia Mare, Romania; 7Faculty of Medicine, Transilvania University, 500019 Brasov, Romania; 8Faculty of Medicine, “Carol Davila” University of Medicine and Pharmacy, 050474 Bucharest, Romania; 9Department of Obstetrics and Gynecology, University Emergency Hospital Bucharest, 050098 Bucharest, Romania

**Keywords:** menopause, hormone replacement therapy, estrogens, progestogens, emerging therapies

## Abstract

Menopause is a natural stage of hormonal aging in women, accompanied by a series of symptoms that reduce the quality of life of a fully active person. As no therapy is entirely satisfactory, the race for a better option is in full swing. Our study objective is to investigate the most recent menopause studies on pharmacological resources, emerging therapies, and the particularities of hormonal replacement therapy (HRT). For this purpose, a comprehensive search was conducted in two main databases (PubMed and Web of Science) guided by the specific keywords “menopause” and “therapy” or “estrogen” or “progesterone” or “hormone replacement” during the last ten years period. Studies were eligible if they met certain criteria: randomized controlled trials (RCT) in adult women with menopause and hormonal or non-hormonal therapies. We selected 62 RCTs, which are focused on four main topics: (a) epidemiology of menopause-related symptoms, (b) hormonal replacement therapy (HRT) selective estrogen receptor modulators, (c) emerging therapies, and (d) menopause. HRT has proven a real health benefit for menopausal women; besides, complementary interventions must be considered. Further studies are needed on menopause and menopause-related therapies. The continuous updating of clinical experience will strengthen the therapeutic benefit and the decision to treat patients safely. This goal will fully access all therapeutic resources to address an unresolved health issue of active adult women.

## 1. Introduction

For many women, menopause marks a decline in hormonal activity, a significant transition in their lives with a more or less dramatic clinical picture. Most menopausal women will have a range of symptoms, accompanied by physiological changes in the reproductive organs and psycho-affective disturbances, with significant adverse implications being a landmark for another life stage. This change is perceived negatively for women’s behavior and is associated with the aging process [1].

From the introduction of the first hormone therapy in 1942 until now, several critical stages have been encountered in the history of hormone therapy. In 2002, the first set of results from the Women’s Health Initiative (WHI) showed a high risk of venous thromboembolism, cardiovascular disease, and breast cancer [2]. Hormone replacement therapy (HRT) significantly changed patient comfort and clinicians’ reluctance, respectively. After a dramatic decline in the use of hormone therapy in menopause, a period that endangered the quality of life of many women, recent years have reconsidered its role in improving menopausal symptoms. HRT mainly acts on vasomotor symptoms, and genitourinary syndrome and prevents fractures [3].

The paradigm of “joint decision-making”—patient-physician, taking both the risks and benefits of menopause hormonal therapy (MHT), must consider the type of hormone use, dose, formulation, administration route, time of initiation, duration of use, and whether the progestin is used. Personalized treatment will consider the benefit-risk balance with periodic reassessment or treatment discontinuation.

An update on the action of HRT is given by the 2017 recommendations of the North American Menopausal Society, which relaxes and encourages the attitude of clinicians regarding the use of hormonal preparations in menopause [4]. The new perspectives on HRT aim to re-examine the latest studies and the proper training of clinicians. Menopause is a significant public health problem requiring strategies to prevent and improve symptoms and reduce complications that frequently occur about ten years after as well as the mortality rate. Therefore, using HRT should aim to prevent chronic diseases and should not be used only to treat vasomotor symptoms [5].

HRT is a serious option for vasomotor and genitourinary symptoms in menopause. Studies by the WHI regarding the need to use hormone therapy evaluated the risk ratio: benefit, safety profile of HRT, time of initiation of the regimen used, and route of administration, related to the patient’s age, menopausal time, and associated conditions. Suitable candidates for HRT are women with moderate or severe symptoms who have recently gone through menopause because the response is superior to postmenopausal women. For a good understanding of the analysis of the positive effects of HRT, a large number of randomized studies and a good interpretation of the results of these studies are needed.

The present review aims to investigate the newest research published regarding (a) the evaluation of the epidemiological data in menopausal women, by reports registered during the last ten years period; (b) recent HRT studies and emerging new medical therapies. Knowing the latest updates on these topics will help medical professionals achieve maximum therapeutic benefits, treat women safely, and access all available menopause therapeutic resources to address the active adult female’s still unsolved health issue.

## 2. Material and Methods (Data Search)

In the present paper, we performed a narrative review searching original published papers on menopause in humans, with a data filter on the quality of life, medical therapy, complementary therapy, and lifestyle support in premenopausal and menopausal women. Other criteria included: published in a scholarly peer-reviewed journal; written in English or French (but with no country restriction); and from the last ten years. The review methods of the search were established beforehand. The initial review protocol assumed a Google Scholar search; due to the document types’ diversity, the search was performed in two major databases, PubMed^®^/MEDLINE, and Web of Science^®^. The report had no other significant deviations from the initial study plan.

**Information sources:** We searched the databases PubMed^®^/MEDLINE and Web of Science^®^ for menopause, quality of life, and related keywords. **Search:** We did a search in the two databases (filters applied: Clinical Trial, Randomized Controlled Trial, Clinical Case Series, on Humans, in the last 10 years) with the keywords: “menopause” AND “quality of life” OR “therapy” OR “treatment.” We restricted searching for articles written in English or French; the last updated search was performed on 25 September 2022.

**Study selection:** Inclusion criteria were: (1) menopausal women, (2) premenopausal women undergoing, (3) hormone replacement therapy, or (4) non-hormonal therapies and being assessed for quality of life (QOL); registered in the clinical trial (CT), randomized clinical trials (RCT) and clinical case series (CCS) published in the last ten years.

**Data extraction:** the following data were selected: author(s), year of publication, country, aim of the study, study design, and main results. Two independent investigators extracted the data and selected a sample of eligible studies, achieving good agreement. Firstly, the authors screened articles by title and abstract and then by full text. We did snowball searches of key papers. Duplicates and articles not fulfilling the search criteria were excluded.

**Data analysis** was performed by three authors (E.C., A.L.P., and B.A.N.). Over 3000 studies with menopause or pre-menopause treated with medicines or non-hormonal therapies, investigated for QOL, with or without a control group, were identified and screened for eligibility by the three examinators. According to the topic search, data extracted included demographic variables, the number of participants in the study, treatment, side effects, and QOL changes. We completed the data collection in September 2021. The quality of the selected studies for review was evaluated based on study type randomized clinical trials (RCT) and clinical trials (CT). A total of 49 papers were included in the present study, centered on the main topics included in the search. The data analysis was performed using Microsoft Excel^®^ 2013 (Microsoft^®^ Corporation, Redmond, WA, USA).

## 3. Menopause Epidemiological Data

Menopause is the enduring physiological cessation of menses for over twelve months as a consequence of estrogen deficiency, without being associated with a pathology. One of the main aims of the World Health Organization (WHO) is to focus on the quality of life of people worldwide. Therefore, the organization has proposed a global strategy focused on aging and health to ensure that adults live a longer and healthier life. Data shows that 25 million women globally enter menopause earlier each year. By calculations, it will result in 1.2 billion postmenopausal women by 2030 throughout the world (WHO, 2020). The mean age for menopause onset in white Caucasian women is 51 years, which is also subject to ethnic and regional variations [6]; in women of Asian origins, menopause occurs at a variable age. For example, in Taiwanese women, the onset of menopause occurs at a mean age of 49.3 years, as indicated by a study conducted in 1997 [7]. Although menopause can occur earlier—either naturally or induced by surgery, radiotherapy, and chemotherapy [8]—approximately 1.3 million women become menopausal each year in the United States.

The menopause phase includes a broad spectrum of psychological, physical, and social changes in a woman’s body, resulting from declining ovarian hormonal activity. Symptoms can alter the quality of life and include mood disorders, sleep difficulties, sexual dysfunction, various degrees of muscle and joint pain, low bone mass, and typical hot flashes.

The spectrum and prevalence of menopause symptoms vary around the world. In South Africa, women express dissatisfaction related to mood disorders, sexual dysfunction, and osteo-muscular pain. In the United States, menopausal symptoms that are highly reported are the ones related to joint and muscle pain. Women in Australia complain mostly about vasomotor symptoms and genitourinary dysfunction, while Asian women report an alarmingly increased incidence of depressive disorders. In the women surveyed in Europe, sleep and depressive disorders were reported more frequently [9]. Individual factors can influence a woman’s perception of menopausal symptoms. Personal variables such as history, current health status, or socioeconomic factors can alter the perception of menopausal onset [10]. Health education empowers women, offering them information about the signs and symptoms of menopause onset and helping them to adopt a more active and aware attitude, therefore improving their quality of life during this life period.

## 4. Menopause Hormone Therapy in Current Practice

HRT represents one of the most efficient treatment options for symptoms in menopause, including symptoms of genitourinary syndrome or vasomotor symptoms [11,12]. Hormonal replacement therapy was highly used in the late 1990s, but prescriptions declined after the publication of a large randomized controlled trial of the Women’s Health Initiative in 2002. The study indicated that for women in post-menopause (63 years, age average), treatment with oral conjugated equine estrogen alone after having a hysterectomy or medroxyprogesterone acetate once a day in women with a uterus—was related to risks exceeding preventive benefits [13,14].

Studies that determined the risk and benefits of hormone replacement therapy had inconsistent results. One recent systematic review and meta-analysis published in 2015, indicates that HRT does not increase all-cause or cause-specific (cardiovascular, stroke, cancer) mortality [15]. Currently, women have a broader spectrum of therapeutic options than ever before. Treatment options may vary in the dose and route of administration; type of product; selective estrogen receptor modulators (SERMs) alone or in combination with other treatments; and a wide spectrum of non-hormonal prescription medications.

### 4.1. Estrogens in Menopause Therapy

#### 4.1.1. Management of Menopausal Genitourinary Symptoms with Estrogen

Sexual dysfunction is frequent in women with genitourinary syndrome of menopause. Thus, vaginal estrogen can be an effective treatment for genitourinary syndrome.

The REJOICE study, published in 2016, showed that low-dose solubilized 17β-estradiol, administered as vaginal soft gel capsules, was safe and potent for treating dyspareunia of moderate-severe symptoms in postmenopausal women complaining of vulvar and vaginal atrophy. The treatment with 4, 10, or 25 μg 17β-estradiol improved Female Sexual Function Index scores gradually, dose-dependent [16].

A randomized, double-blind, multicenter study (n = 576) showed similar benefits of vaginal low-dose estradiol cream. Postmenopausal women with vaginal dryness—the most incommodious genitourinary symptoms were randomized to estradiol cream 0.003% (15 μg estradiol; 0.5 g cream) or placebo. Estradiol vaginal cream, applied twice a week, proved an efficient and well-tolerated treatment for vulvar and vaginal atrophy symptoms associated with menopause [17]. A 12-week topical treatment with testosterone (300 μg testosterone propionate), conjugated estrogen, or polyacrylic acid was superior to placebo lubrication. In addition, the regimen significantly improves vaginal trophic in postmenopausal women complaining of vaginal atrophy, as shown in a randomized clinical trial conducted at the Menopause Clinic of CAISM UNICAMP between December 2011 and January 2013 [18].

A multicenter, randomized trial (The VeLVET Trial) compared vaginal laser therapy to vaginal estrogen cream in women with significant symptoms of vaginal atrophy. After six months, both treatments showed similar improvement in urinary and sexual function (n = 69) [19,20]. A double-blind trial including postmenopausal women with sexual dysfunction studied the effects of a conjugated estrogens tablet (0.625 mg) for intravaginal administration (n = 67). The study showed no significant changes in the Female Sexual Function Index. However, a significant improvement in the vaginal pH and Vaginal Maturation Value was observed [21].

Thus, there are several therapeutic schemes to improve genitourinary symptoms. Current treatment is based on topical estrogen alone [21,22,23,24,25,26] or combined with testosterone [27,28] which has shown improvements over time in the fields of vaginal trophicity, arousal, orgasm, and satisfaction (Table 1). For moderate/severe dyspareunia the treatment uses daily intravaginal 0.50% DHEA ovules (suppositories—Prasterone) for 12 weeks [29,30]. Thus, in postmenopausal patients with dyspareunia, oral administration of ospemifene 60 mg daily may be performed for 12 weeks [31,32] or 58 weeks [33]. Another combination of estrogen therapy uses the intravaginal fractional CO2 laser for treating symptoms related to genito-urinary syndrome [19,20,34,35]. The combined estrogen-progesterone scheme decreased the rate of urogenital atrophy symptoms and the frequency and severity of hot flushes [36]. Another study shows that vitamin E may be an alternative to vaginal estrogen [26].

#### 4.1.2. Management of Cardiovascular Diseases in Menopause with Estrogen

Endocrine and metabolic changes that occur in the transition to menopause may accelerate the risk of cardiovascular disease induced by high blood pressure. According to the American Heart Association, the initiation of hormone therapy for menopause in women under 60 years of age or within 10 years of menopause is the only prophylactic method to reduce the risk of cardiovascular disease [37].

A multicenter, randomized placebo-controlled trial (KEEPS) evaluated estrogen therapy’s effect on cardiac adipose tissue accumulation and the link with calcification of coronary arteries. Participants (n = 727) received either oral-conjugated equine estrogen (o-CEE, 0.45 mg/d), transdermal estrogens 17b-estradiol (t-E2; 50 mcg/d), or placebo (inactive pill and patch). Participants who received o-CEE or t-E2 were also given oral micronized progesterone (200 mg/d) for the first 12 days every month. The study showed that in women who recently entered menopause, treatment with oral conjugated equine estrogens might slow epicardial adipose tissue accumulation. In contrast, transdermal 17β-estradiol can increase coronary artery calcification progression associated with paracardial adipose tissue accumulation [38].

A randomized, double-blind place-controlled trial on transdermal estradiol investigated the short-term effects of this compound as a low-dose therapy on the endothelium and its function, insulin sensitivity in nondiabetic women with a body mass index above 25 (overweight or obese) at menopause. Participants (n = 44) received estradiol hemihydrate gel or placebo gel for three months. Transdermal estradiol administered in a low dose for a short period proved effective, showing a beneficial effect on endothelial function and reducing blood viscosity compared to placebo [39].

In a double-blinded, placebo-controlled trial, 64 healthy postmenopausal women were randomly allocated to 0.625 mg conjugated equine estrogens or a placebo for 28 days. The study showed that the administration of 0.625 mg conjugated equine estrogens effectively improves vascular nitric oxide-dependent dilation, assessed by flow-mediated dilation of the brachial artery [40].

Although data indicate that HRT is associated with positive effects on the cardiovascular system in women in menopause, it is still a question whether these effects vary with the timing of treatment initiation. In a large randomized controlled study documenting the relationship between timing and effects of HRT on the cardiovascular system in menopausal women, 643 healthy women in menopause randomly received oral 17 β-estradiol (1 mg daily) or a matching placebo. Women who had a uterus and were in the estradiol group were also given micronized progesterone (45 mg) as a 4% vaginal gel, and women with the uterus in the placebo group were assigned a matching placebo gel; the estradiol or the placebo gel had to be applied in a sequential matter (once a day for ten days during every 30-day cycle). The study showed that HRT with oral estradiol was associated with less subclinical atherosclerosis progression (measured as carotid-artery intima-media thickness) than placebo when the therapy was initiated up to six years after a woman entered the menopause phase, not immediately after onset or more than ten years later. In addition, atherosclerosis deposits evaluated by cardiac CT did not vary significantly with estradiol therapy in both menopause groups [41].

A double-blind, placebo-controlled, randomized trial investigated the quality of life in perimenopausal and menopausal women dealing with vasomotor symptoms (n = 339). The two therapies compared to placebo included 17β-estradiol administered orally in a low-dose manner (0.5 mg/day) and venlafaxine XR 75 mg/day. Analyses showed that low-dose estradiol therapy had beneficial effects on all satisfaction scores included in the questionnaire in women in menopause with vasomotor symptoms, except in the psychosocial domain [42]. Postmenopausal use of cardiovascular disease (CVD) risk scores (the Framingham Heart Study BMI score) select women at higher risk of CVD and help better advise on the risks/benefits of HRT before initiating hormone therapy [43]. WHI’s findings indicate that systemic hypertension has a good safety profile when HRT was initiated in healthy women with newly installed menopause without an increased risk of cardiovascular disease or breast cancer, compared to postmenopausal women [44].

Therapeutic regimens for the treatment of hot flashes include only progesterone [45], the combination of estrogen–progesterone [46,47], estrogen–combined selective serotonine reuptake inhibitors (SSRI) [42,48], SSRI [49], or anticonvulsant agents [50,51] (Table 2). For moderate/severe vasomotor symptoms, neurokinin 3 receptor antagonists [52,53,54], estrogen combination—SERM [55], noradrenergic and specific serotonergic antidepressant (NaSSA) [56] or antipsychotics [57] are used.

#### 4.1.3. Management of Menopausal Neuropsychological Symptoms with Estrogen

The ELITE-Cog (Class I evidence) trial investigated whether estradiol therapy initiated in the first six years of menopause has a different impact on verbal than when initiated ten or more years later. Participants were given oral 17β-estradiol (1 mg/d) or a placebo at a single academic medical center site (University of Southern California). In addition, women with a uterus were also assigned to cyclic micronized progesterone (45 mg) as a 4% vaginal gel or placebo gel, one daily application for ten days per 30-day cycle. The trial showed that estradiol initiated within six years of menopause does not affect verbal memory, executive functions, or global cognition in a different matter than treatment that began ten or more years after menopause. Furthermore, estradiol does not improve or harm these cognitive abilities regardless of the time since menopause [61].

Similar results were observed in a trial, including 59 hysterectomized, middle-aged postmenopausal women who received either conjugated equine estrogens of 0.625 mg/day or placebo during the six rounds comprising 28 days. Estrogen replacement therapy did not improve verbal memory in hysterectomized, asymptomatic postmenopausal women [62].

A randomized controlled study, including 76 hysterectomized menopausal women, investigated whether estrogen replacement therapy does improve mood and anxiety symptoms in non-depressive women at menopause. Women received conjugated equine estrogen tablets (0.625 mg/day) or a matching placebo. Treatment was administered once a day at bedtime for 28 days without intervals, after which a new cycle began, for a total of six cycles. The study revealed that estrogen replacement therapy does not improve mood or anxiety levels in non-depressive, hysterectomized, healthy postmenopausal women [63]. In patients under 80 years of age who used estrogen alone for >10 years, a lower overall risk of dementia was observed, while the use of estrogen-progestin therapy was associated with a potentially increased risk of developing Alzheimer’s disease [64].

Therapeutic regimens for the treatment of sleep disorders include medroxyprogesterone acetate MPA [65], selective serotonin and norepinephrine reuptake inhibitors (SSNRI) (venlafaxine)/SSRI (citalopram) [66], estrogen and antidepressants (SNRRI) [58] or SSRIs only [59,60] or an estrogen-progesterone combination [67] and for cognitive fatigue estrogen–testosterone combination [68] (Table 3). HRT has good results on the subjective assessment of sleep, but not the objective one by measuring sleep parameters by polysomnography. Thus, 17β-estradiol improves sleep quality, while the combination of estrogen and progesterone acts positively on sleep disorders. Transdermal administration is more effective than oral [69]. For major depressive disorders are indicated SSRIs [70], SERM (tibolone) [71], or antipsychotics (lurazidone) [72].

#### 4.1.4. Management of Menopausal Musculoskeletal Symptoms with Estrogen

The Women’s Health Initiative randomized, placebo-controlled trial, including 10,739 women in the menopause phase with a history of hysterectomy, examined the effects of estrogen replacement therapy on various menopausal symptoms, including joint symptoms. Patients were randomly assigned either oral conjugated equine estrogens (0.625 mg/day) or a matching placebo. The trial showed that hormonal replacement with estrogen alone results in a modest but sustained scaling down in joint pain frequency in postmenopausal women [74] (Table 4).

The KEEPS trial investigated the effect of estrogen replacement therapy on cortical and trabecular bone. Patients were assigned 0.45 mg/day of oral conjugated equine estrogens, 50 mcg/day of transdermal 17-βestradiol (both with oral, micronized progesterone, 200 mg for 12 days each month), or placebo. The trial results indicate that hormonal replacement therapy has site-specific effects on cortical versus trabecular bone that must be investigated further [76].

### 4.2. Progestogens in Menopause Therapy

Progesterone is usually administered in combination with estrogen in hormonal replacement treatment. Few studies have investigated the effects of progesterone alone on menopausal symptoms. Most of these studies focused on the effects of progestogens on sleep quality. A randomized, double-blind cross-over study tested the effects of intranasal progesterone, zolpidem, and placebo on healthy women’s sleep in menopause. In a randomized order, women were assigned to receive four treatments: two intranasal progesterone puffs (4.5 mg and 9 mg of MPP22), 10 mg of zolpidem, and a placebo. The study indicated that intranasal progesterone, administered to healthy postmenopausal women, is beneficial for sleep [77].

Similar improvements in sleep quality have also been observed in a trial investigating how different progestogens affect sleep in postmenopausal women. Participants were randomly assigned to two separate groups in a trial performed by the Menopause Clinic at Maharaj Nakirn Chiang Mai Hospital in Thailand, including 100 Thai women dealing with insomnia. Both groups received estradiol valerate (1 mg) daily. Dydrogesterone (10 mg) was assigned to the first group, and micronized progesterone (100 mg) was given to the second group. After the first month of treatment, sleep quality improved dramatically in both groups, and fewer adverse effects were registered in the groups receiving dydrogesterone [67].

### 4.3. Selective Estrogen Receptor Modulators (SERMs) in Menopause Therapy

#### 4.3.1. Management of Menopausal Symptoms with Ospemifene

Ospemifene is an FDA-approved oral selective estrogen receptor agonist/antagonist. Indications for using ospemifene are moderate or severe dyspareunia, and the drug was recently approved for moderate/severe vaginal dryness, which is a frequent symptom of vulvovaginal atrophy [78].

A 12-week phase three study investigated the beneficial effects of ospemifene on postmenopausal women’s vulvovaginal symptoms (n = 631). Participants were randomly assigned to either daily ospemifene 60 mg or placebo, assessing effects using prospective vulvar-vestibular photography and observation. The study revealed that ospemifene improves vulvovaginal health and efficiently manages moderate/severe vaginal dryness [32].

Other studies reported similar benefits of ospemifene use. For example, a phase three, randomized, double-blind 12-week trial (n = 919) included postmenopausal women reporting dyspareunia or vaginal dryness as the most bothersome symptom. Participants received either oral ospemifene 60 mg/day or a placebo for 12 weeks. The study concluded that ospemifene significantly improves the Female Sexual Function Index in women complaining of vaginal atrophy [31].

Another phase three clinical trial compared the improvement in vulvovaginal symptoms between ospemifene or lubricants. Subjects received ospemifene 60 mg/day or a placebo for 12 weeks. Compared with baseline, patients assigned to ospemifene had a complete resolution of clinical signs of vulvovaginal atrophy after 12 weeks and benefits sustained for 52 weeks. Thus, the study concluded that ospemifene has a beneficial impact on vulvovaginal symptoms and substantially improves clinical signs of vulvovaginal atrophy [79].

#### 4.3.2. Management of Menopausal Symptoms with Conjugated Estrogens/Bazedoxifene

Conjugated estrogens/bazedoxifene (CE/BZA) is the first medication approved by the FDA, including conjugated estrogens with an estrogen agonist/antagonist (bazedoxifene) [80]. Various trials have investigated how CE/BZA impacts both quality of life and health, showing that the drug complex has beneficial effects on various menopause-related symptoms.

The SMART-1 and SMART-2 trials evaluated the effect of CE/BZA on various symptoms in menopausal women. Subjects were randomly assigned either bazedoxifene 20 mg/conjugated estrogens 0.45 or 0.625 mg (CE/BZA) or placebo. The trials primarily investigated the frequency and intensity of hot flushes, but the quality of life, sleep, anorexia, and other menopause-related symptoms were also targeted. The study showed that the CE/BZA complex significantly decreased the severity and frequency of hot flushes than the placebo and improved health-related quality of life [81].

The effect of CE/BZA on sleep and health-related quality of life was also investigated by another phase three trial, including 459 women with moderate/severe vasomotor symptoms. Patients were randomized to BZA 20 mg/CE 0.45 mg, BZA 20 mg/CE 0.625 mg, BZA 20 mg, CE 0.45 mg/medroxyprogesterone acetate (MPA) 1.5 mg, or placebo for one year. Sleep parameters and health-related quality of life were assessed at months 3 and 12. CE/BZA and CE/MPA had a beneficial effect, improving sleep and health-related quality of life in symptomatic postmenopausal women with bothersome vasomotor symptoms [82].

The SMART-5 trial, which evaluated the positive effects of CE/BZA in postmenopausal women with bothersome symptoms, indicated that the drug complex does not have adverse effects on lipid metabolism and hemostatic balance. Venous thromboembolic or cardiovascular events occurred in a similar incidence as in the placebo group [83].

## 5. Non-Hormonal Medical Therapy

### 5.1. Management of Menopausal Symptoms with Bisphosphonates

Bisphosphonates (BP) are key pharmaceutical agents acting against osteoclast-mediated bone loss. Thus, the most frequent medical condition for which BP are used is osteoporosis in menopause—a pathology that in Romania has been shown to appear ten years earlier than the internationally accepted limit [84]. In postmenopausal osteoporosis, there is an imbalance between bone resorption mediated by osteoclasts and bone formation mediated by osteoblasts, such that the former is increased; hypercholesterolemia accelerates postmenopausal bone loss [85]. On the other hand, BP inhibits osteoclast activity, therefore leading to a greater bone mineral density, reducing bone turnover, replenishing the resorption spaces, and mineralizing the extracellular matrix [86].

Most studies were focused on safety, compliance with treatment, and pharmacogenetic aspects of BP therapy in menopausal women. In a double-blind placebo-controlled RCT study on 324 postmenopausal women; 161 younger and 163 older women were randomized to 35 mg/week risedronate I or placebo. The study showed that risedronate reduced serum carboxyterminal cross-linking telopeptide of type one bone collagen (CTX-1) and serum amino-terminal propeptide of type one procollagen (P1NP) by approximately 50%. Risedronate slowed microstructural deterioration in younger and partly reversed it in older postmenopausal women, which may decrease the fracture risks in menopausal women [87]. This feature was pointed out previously on a 2010 RCT comparing alendronate (A) to risedronate I.

Alendronate produced significantly greater mean bone mineral density (BMD) increases from baseline than risedronate at the hip trochanter, lumbar spine, total hip, and femoral neck [88]. However, one of the most frequent negative effects of BPs is osteonecrosis of the jaw, which occurred more often in menopausal women treated with Alendronate [81]; the decreased bone formation after several months of BP was associated with increased serum levels of Wnt signaling antagonist sclerostin, the protein secreted by mature osteocytes during the completion of osteon formation that inhibits bone formation [89]. BPs are anti-resorptive medicines that have been proven to be effective in preventing bone loss and vertebral fractures for postmenopausal women with very low bone mineral density and/or those who have had previous fractures [90]. However, a potentially severe complication such as atypical subtrochanteric or femoral shaft fractures and Medication-Related Osteonecrosis of Jaws (MRONJ) is more frequently observed in metastatic bone cancer and myeloma patients receiving BP agents, but it is also diagnosed in patients receiving BPs for postmenopausal osteoporosis [91]. The awareness of side effects has increased since 2004, but knowledge about risk assessment and management is still insufficient [92].

Conversely, 12 months study on 22 osteopenic osteoporotic women showed that BP therapy, when combined with scaling and root planning, is associated with positive outcomes in postmenopausal women dealing with moderate-severe chronic periodontitis [93].

A triple-blind RCT study on 60 women investigated the risk of developing diabetes mellitus in prediabetic patients and showed that alendronate (70 mg/week) improved fasting plasma glucose, HbA1c, and insulin indices in postmenopausal women [94].

Nevertheless, the beneficial effects are endangered by a lack of therapeutic compliance [95] in menopausal women [96,97] confirmed by a cohort study in the UK on 72,256 menopause or over 50 years women, showing that just one in three patients who received osteoporosis therapy continued to be on treatment after two years [98]. A multicenter retrospective observational study on 387 women diagnosed with postmenopausal osteoporosis and with BP treatment showed that the oral soluble form of BPs (A) was better tolerated than the solid oral form of the drug (83.3% vs. 66.7%); comorbidities, polipragmasia, not taking the vitamin D and surgically induced menopause were positively influencing the discontinuation of BP [99,100].

A study on 201 women with BPs therapy showed a genetic marker to treatment outcomes. Single gene variants and their allelic combinations (SOST rs1234612 T/T, PTH rs7125774 T/T, FDPS rs2297480 G/G, and GGPS1 rs10925503 C/C+C/T) influence the pharmacogenetics of BPs in the treatment of osteoporosis, being overexpressed in the BP non-responders group; in the future, genetic markers screening could be implemented to determine which individuals might benefit most from the antiresorptive therapy [101].

### 5.2. Management of Menopausal Symptoms with Kisspeptins

Kisspeptins regulate the release of reproductive hormones, peptides encoded by the KISS1 gene [102], a key regulator of hypothalamic gonadotropin-releasing hormone (GnRH) neurons; their discovery conducted to the investigation of the role of kisspeptin in reproductive disorders and of the potential of targeting the kisspeptin system (KISS1R) therapeutically [103]. At the moment, the investigational GnRH/gonadotropins are Fezolinetant, MVT-602, Linzagolix.

Fezolinetant is an oral neurokinin three receptor antagonist, which selectively and reversibly blocks neurokinin B signaling. Antagonism of neurokinin B contributes to decreased kisspeptin/neurokinin B/dynorphin neuron activity, hence modifying signaling in the thermoregulatory center in the hypothalamus and influencing vasomotor symptoms. Fezolinetant is currently in clinical development and could offer an alternative mechanism to hormonal replacement therapy to control vasomotor symptoms. A phase 2a trial evaluated the safety and efficacy of fezolinetant in women with vasomotor symptoms (n = 80). Patients were assigned to either a placebo or 90 mg of fezolinetant twice daily, packed as a capsule, and administered with a light meal approximately 12 h apart for 12 weeks. The study showed that the intensity of moderate or severe vasomotor symptoms was reduced from the beginning, and benefits were registered in all QoL domains. The drug had good tolerance. The results of this Phase 2a trial concluded that fezolinetant has rapid action and the potential to determine a significant reduction in moderate-severe hot flushes. Therefore, the drug could be a valuable non-hormonal therapeutic option for postmenopausal women [104].

MVT-602 (RVT-602, TAK-448) is a small-molecule kisspeptin receptor agonist for female infertility and hypogonadism [105].

Linzagolix (KLH-2109, OBE-2109)—small-molecule GnRH receptor antagonist for uterine fibroids and endometriosis (PRIMROSE, respectively, Edelweiss trials) [106,107].

## 6. Discussion

The age of menopause is a time of many changes in women’s psychosocial and functioning, with reduced ovarian hormonal activity and estrogen levels and disturbing transition symptoms. The most common menopause symptoms include mood disorders, sleep difficulties, sexual dysfunction, various degrees of muscle and joint pain, low bone mass, and typical hot flashes.

The prevalence of menopause symptoms is different worldwide, with various dominant complaints—pain, and joint aches in Asia and USA; depressive, sexual dysfunctions and pains in South America; vasomotor symptoms and sexual dysfunction in Australia; depressive disorders in Asia, associated with sleep disorders in Europe. In addition, menopause decreases the health-related quality of life (HRQoL) dependent on work and other sociodemographic variables.

HRT represents one of the most efficient treatment options for symptoms in menopause but continues to be controversial regarding its use. HRT and selective estrogen-receptor modulators should not be used for the prevention of cardiovascular disease (CVD) [108]; if prescribed, it must remain a short-term solution, using the lowest dose possible to minimize night sweats, the systemic use being reserved for disabling night sweats and hot flashes and not for long term primary and secondary CVD prevention [109].

In the last two decades, the average age at natural menopause and the average reproductive life increased by 0.5 years, the body mass index (BMI) increased by 1.1, the age of onset of hypertension increased by 5.1 years and the use of hormone therapy decreased by 20.4 years [110]. Increasing the average age at natural menopause is associated with decreased mortality from any cause, especially cardiovascular, although high blood pressure starts earlier. Other studies showed that late menopause reduced the risk of cardiovascular disease, osteoporosis, and dementia [111]. Furthermore, Ley et al. revealed that a shorter period of reproductive life span increased the risk of CVD [112]. Earlier use of hormone therapy may also reduce cardiovascular risk, but increase the risk of endometrial, breast, and ovarian cancer.

From the presentation of the results of the women’s health initiative in 2002 to the end of 2009, the demand for hormone therapy has been reduced by more than 70%, while the demand for low-dose transdermal hormone therapy has increased 10-fold [113]. Another study showed that during the same period, vaginal and oral preparations with lower doses were preferred [114].

In our study, we highlighted the main contributions of HRT on menopause symptoms; a conjugated estrogens tablet for intravaginal administration significantly improved the vaginal pH and Vaginal Maturation Value, with no significant changes in the Female Sexual Function Index (FSFI). Low-dose solubilized 17β-estradiol, administered as vaginal soft gel capsules, was safe and potent for treating dyspareunia of moderate-severe symptoms in postmenopausal women complaining of atrophic vaginitis. Estradiol vaginal cream, applied two times a week, proved to be an efficient and well-tolerated treatment for vulvar and vaginal atrophy symptoms associated with menopause. Topical treatment with testosterone, conjugated estrogen, or polyacrylic acid was superior to placebo lubrification, significantly improving vaginal trophic in postmenopausal women with vaginal atrophy.

Vaginal laser therapy and vaginal estrogen cream showed similar improvement in urinary and sexual function. In a study of 85 randomized postmenopausal women, Li et al., no significant improvement was observed between laser treatment with fractionated carbon dioxide versus simulated treatment of vaginal symptoms after 12 months [115]. Although there is a risk of tissue damage [116] the use of fractional microablative CO2 laser therapy in women who have survived gynecologic cancers is an effective procedure in the management of vulvovaginal atrophy, given that estrogen therapy is contraindicated [117].

If initiated within six years of menopause, estradiol does not affect verbal memory, executive functions, or global cognition in a different matter than treatment that began after the first ten years of menopause. Estradiol does not benefit or harm these cognitive abilities regardless of menopause. Estrogen replacement therapy (ERT) did not improve verbal memory in hysterectomized, asymptomatic postmenopausal women. ERT does not improve mood or anxiety levels in non-depressive, hysterectomized, otherwise healthy postmenopausal women. However, hormonal replacement with estrogen alone results in a modest but sustained scaling down in joint pain frequency.

In women who recently entered menopause, oral conjugated equine estrogens might have a beneficial influence on the rhythm of epicardial fat accumulation. In contrast, transdermal 17β-estradiol can lead to a more rapid coronary arteries progression linked to paracardial fat deposits. Low-dose estradiol therapy had beneficial effects on all satisfaction scores included in the questionnaire in women at menopause with vasomotor symptoms, except the psychosocial domain.

Hormonal therapy with oral estradiol was associated with minor subclinical atherosclerosis progression (measured as carotid-artery intima-media thickness) than placebo when treatment was started in the first six years after menopause onset after ten or more years since the onset of menopause. Conjugated equine estrogens effectively improve vascular nitric oxide-dependent dilation in healthy postmenopausal women. Estrogen with micronized progesterone has site-specific effects on cortical versus trabecular bone that need to be further investigated. In nondiabetic overweight or obese women who recently entered menopause, transdermal estradiol administered in a low dose for a short period proved to be effective, showing a beneficial effect on endothelial function and reducing blood viscosity in comparison with placebo.

Intranasal progesterone, administered to healthy postmenopausal women, has beneficial effects on sleep. Sleep quality dramatically improves postmenopausal women with insomnia after the first month of treatment with estradiol valerate and either dydrogesterone or micronized progesterone.

Ospemifene improves vulvovaginal health and is efficient in managing moderate-severe vaginal dryness. The drug significantly improves the Female Sexual Function Index in women after the onset of menopause with genitourinary symptoms. Bazedoxifene-conjugated estrogens complex substantially decreases the severity and frequency of hot flushes compared to placebo and improves health-related quality of life scores. Bazedoxifene-conjugated estrogens complex does not have adverse effects on lipid metabolism and hemostatic balance. Fezolinetant has a rapid action and can determine a significant reduction in moderate-severe hot flushes. Therefore, the drug could be a valuable non-hormonal therapeutic option for postmenopausal women.

Ovarian tissue banking has been proposed as a new feasible procedure to delay menopause by endocrine function restoration, not only for fertility preservation [118,119].

BP therapy is beneficial in MP-associated osteoporosis and may improve fasting plasma glucose, HbA1c, and insulin indices in postmenopausal women; however, the treatment compliance is still reduced, and efficacy may be genetically influenced by non-responders to therapy. Screening of BP pharmacogenetic markers may improve antiresorptive therapy in menopausal and postmenopausal women [120].

## 7. Conclusions

Menopause is a physiological period during a woman’s lifetime, accompanied by a wide spectrum of symptoms that can alter the quality of life of a fully active adult; HT use, the most valuable therapeutic resource to increase QoL in menopausal women, is limited to moderate and severe symptoms. Furthermore, many women choose not to use MHT due to the fear of cancer or other negative effects. Unfortunately, no entirely satisfactory therapy is available, so the future search for better or complementary options is essential. Knowing the latest updates on these topics will help medical professionals better achieve the therapy benefit, treat women within safety margins, and to access all available menopause therapeutic resources to address the active adult female’s still unsolved health issue.

Women in early menopause with present symptoms can benefit from HT in safe conditions and with clinically proven effectiveness, allowing the improvement of the quality of life. An important role in menopause regarding the therapeutic strategy is represented by the trust in the administration of hormonal therapy both among patients and among doctors.

## Figures and Tables

**Table 1 life-13-00649-t001:** Synopsis of the main studies of urogenital symptoms.

Authors, Year	Cases	Symptoms	Treatment	Category	Effect
Simon, 2013 [33]	826	Vulvovaginal atrophy	Oral ospemifene 60 mg/day for 52 weeks. Participants either continued their 60-mg/day ospemifene dose from the initial 12-week pivotal efficacy study or switched from blinded placebo or ospemifene 30 mg/day to open-label ospemifene 60 mg/day. The 52-week open-label extension period plus initial 12-week treatment period totaled up to 64 weeks of ospemifene exposure. A 4-week posttreatment follow-up ensued (68 weeks total).	Oral selective estrogen receptor agonist/antagonist (Ospemifene)	Ospemifene is clinically safe and generally well tolerated in postmenopausal patients with dyspareunia, a symptom of VVA.
Fernandes, 2014 [27]	80	Low female libido	Vaginal application three times a week for 12 weeks: Vaginal cream with polyacrylic acid: one vaginal applicator with 3 g cream per application. Vaginal cream with testosterone propionate: one vaginal applicator with 1 g of cream per application containing 300 μg testosterone propionate prepared using testosterone micronized powder in an emollient cream with silicone. Vaginal cream with conjugated estrogens: one vaginal applicator with 1 g of cream per application containing 0.625 mg conjugated estrogens. Lubricant with glycerin gel: 3 g in one applicator per application adjusted to maintain similarity with the polyacrylic acid application. (control group).	MHT (Estrogen, Testosterone)	Treatment with topical estrogen in comparison with lubricant alone showed an improvement in the field of desire. The intragroup analysis over the time of the treatment showed improvements in the fields of desire, lubrication, and reduced pain for polyacrylic acid, testosterone, and estrogen. Women who used testosterone showed improvements over time in the fields of arousal, orgasm, and satisfaction.
Constantine, 2015 [31]	919	Dyspareunia	Oral ospemifene 60 mg/day vs. placebo for 12 weeks.	Oral selective estrogen receptor agonist/antagonist (Ospemifene)	Ospemifene 60 mg/day significantly improved female sexual disfunction in women with vulvovaginal atrophy.
Archer, 2015 [29]	253	Moderate/severe dyspareunia	12-week daily intravaginal administration of 0.25% (3.25 mg) DHEA and 0.50% (6.5 mg) DHEA suppositories, compared with placebo.	MHT (DHEA-Prasterone)	Daily intravaginal Prasterone (0.50%; 6.5 mg) has clinically and significant beneficial effects on vulvovaginal atrophy. No significant drug-related adverse effect in line with the strictly local action of treatment has been reported.
Cortés-Bonilla, 2015 [36]	103	Urogenital atrophy and hot flushes	17β-estradiol I/progesterone (P)—IM application of -monthly intramuscular injection of 0.5 mg E + 15 mg P, 1 mg E + 20 mg P or 1 mg E + 30 mg P for 6 months.	MHT (Estrogen, Progesterone)	The three treatment schemes significantly decreased the rate of urogenital atrophy symptoms and the frequency and severity of hot flushes.
Fernandes, 2016 [28]	80	Vaginal atrophy	One of four treatment groups, three times a week over 12 weeks: - Vaginal cream containing 3 g polyacrylic acid; vaginal cream with 300 µg testosterone propionate; vaginal cream with 0.625 mg conjugated estrogens; lubricant with 3 g glycerin gel.	MHT (Estrogen, Testosterone) Polyacrilic acid	Treatment with testosterone and estrogen compared with placebo lubrication, resulted in significant improvement in vaginal trophism in postmenopausal women with vaginal atrophy.
Kingsberg, 2016 [22]	764	Moderate/severe dyspareunia	TX-004HR (4, 10, or 25 µg) (vaginal soft gel capsule containing low-dose solubilized 17β-estradiol) with placebo for 12 weeks.	MHT (estradiol)	TX-004HR dose-dependently improved sexual function, with 25 µg having the greatest effect compared with placebo.
Labrie, 2018 [30]	482	Moderate/severe dyspareunia or pain at sexual activity	Daily intravaginal 0.50% DHEA ovules (suppositories) (6.5 mg) (Prasterone) or matching placebo for 12 weeks.	MHT (DHEA)	Daily intravaginal administration of 0.50% (6.5 mg) DHEA (Prasterone) shows clinically and highly significant effects.
Diem, 2018 [23]	302	Moderate/severe vulvovaginal symptoms	Vaginal 10 mg estradiol tablet plus placebo gel (n = 102), vaginal moisturizer plus placebo tablet (n = 100), or dual placebo (n = 100).	MHT (Estrogen) Moisturizer	Low-dose vaginal estradiol, but not vaginal moisturizer, modestly improved menopause-related QoL and sexual function domain scores.

**Table 2 life-13-00649-t002:** Synopsis of the main studies of the vasomotor symptoms.

Authors, Year	Cases	Symptoms	Treatment	Category	Effect
Hitchcock, 2012 [45]	133	Hot flushes and night sweats	Oral micronized progesterone in a dose of 300 mg (as three 100-mg capsules) or 3 identical placebos taken daily at bedtime for 12 weeks.	MHT (Progesterone)	Oral micronized progesterone is effective for treatment of hot flushes and night sweats in healthy women early in postmenopause.
Simon, 2019 [46]	647	Moderate/severe vasomotor symptoms	TX-001HR (17b-estradiol [E2] and natural progesterone [P4] in a single oral capsule) versus placebo.	MHT (Estrogen, Progesterone)	TX-001HR significantly improved menopause-related quality of life, reducing moderate to severe VMS.
Savolainen-Peltonen, 2014 [47]	150	Hot flushes	Transdermal estradiol hemihydrate gel 1 mg/day (TE). Oral estradiol valerate 2 mg/day (OE) OE with medroxyprogesterone acetate (MPA) 5 mg/day or placebo for 6 months.	MHT (Estrogen, Progesterone)	Estradiol therapy improves sleep, anxiety and fears, and memory in relation to alleviation of hot flashes. The addition of MPA does not negate the effects of estradiol on early postmenopausal women.
Caan, 2015 [42]	339	Hot flushes	Low-dose oral 17beta-estradiol 0.5 mg/day and venlafaxine XR 75 mg/day or placebo for 8 weeks	MHT (Estrogen) SSRI (Venlafaxine)	Both low-dose E2 and venlafaxine are effective for improving QoL in healthy women with VMS.
Joffe, 2014 [48]	339	Vasomotor symptoms	Oral 17β-estradiol (0.5 mg/d), low-dose venlafaxine hydrochloride extended release (75 mg/d), and placebo for 8 weeks. Those randomized to venlafaxine hydro-chloride were titrated from 37.5 up to 75 mg/d during a 1-week period.	MHT (Estrogen) Serotonin-norepinephrine reuptake inhibitor (Venlafaxine)	Low-dose oral estradiol and venlafaxine are effective treatments for VMS. While the efficacy of low-dose estradiol may be slightly superior to that of venlafaxine, the difference is small and of uncertain clinical relevance.
LaCroix, 2012 [49]	205	Vasomotor symptoms	Escitalopram 10–20 mg/day for 8 weeks vs. placebo.	Antidepressant of the selective serotonin reup-take inhibitor (SSRI) class	Treatment with escitalopram 10–20 mg/day in healthy women with vasomotor symptoms significantly improved menopause-related quality of life and pain.
Saadati, 2013 [50]	60	Hot flushes	300 mg gabapentin three times a day for three months, while control group received a placebo.	Anticonvulsivant	The use of gabapentin could decrease the intensity, duration, and frequency of hot flashes in postmenopausal women.
Pinkerton, 2014 [51]	600	Hot flushes	Gastroretentive gabapentin (G-GR) (600 mg am/1200 mg pm) and placebo for 23 weeks	Anticonvulsivant	G-GR has been found to be modestly effective at reducing the frequency and severity of hot flashes for up to 24 weeks, but it does not meet the FDA criterion of a reduction of 2 or more hot flashes/day.
Depypere, 2019 [52]	80	Moderate/severe vasomotor symptoms	90 mg of fezolinetant twice daily or placebo for 12 weeks	Neurokinin 3 Receptor Antagonist	Fezolinetant rapidly and significantly reduced moderate/severe vasomotor symptoms. Potential as an effective nonhormonal treatment option for menopausal women.
Santoro, 2020 [53]	352	Moderate/severe vasomotor symptoms	One of seven fezolinetant dosing regimens (15, 30, 60, or 90 mg BID or 30, 60, or 120 mg QD) or placebo for 12 weeks.	Neurokinin 3 Receptor Antagonist	Oral fezolinetant was associated with higher responder rates than placebo and larger improvements in quality of life and other patient-reported outcome measures, including a reduction in VMS-related interference with daily life.
Fraser, 2020 [54]	352	Moderate/severe vasomotor symptoms	Fezolinetant 15, 30, 60, or 90 mg BID, or fezo-linetant 30, 60, or 120 mg QD (taken in the morning, accompanied by placebo taken in the evening to maintain the blind), or placebo BID for 12 weeks.	Neurokinin 3 receptor antagonist	Fezolinetant is a well-tolerated, effective nonhormone therapy that rapidly reduces moderate/severe menopausal VMS.
Pinkerton, 2017 [55]	318	Moderate/severe hot flushes	Once-daily conjugated estrogens/bazedoxifene 0.45 mg/20 mg, 0.625 mg/20 mg, or placebo for 12 weeks.	MHT (Estrogen) SERM	Not all women using conjugated estrogens/bazedoxifene achieve permanent elimination of hot flushes, the frequency is likely to be substantially reduced during the first week to month. Women can expect approximately 50% reduction in hot flush frequency after 8 to 10 days, and sustained improvement with continued treatment.
Birkhaeuser, 2019 [56]	1888	Moderate/severe vasomotor symptoms	One of five double-blinded treatment groups (placebo, esmirtazapine 2.25 mg, 4.5 mg, 9.0 mg, or 18.0 mg)	Noradrenergic and specific serotonergic antidepressant (NaSSA)—Esmirtazapine	Esmirtazapine significantly reduced the average daily frequency of moderate to severe VMS
Borba, 2019 [57]	28	Moderate/severe hot flushes	Sulpiride 50mg/day or placebo for 8 weeks	Antipsychotic	Sulpiride significantly reduced the total weekly mean of hot flashes and severity scores.
Ensrud, 2015 [58]	339	Vasomotor symptoms (insomnia and sleep quality in women with hot flushes)	17β estradiol 0.5 mg/day, venlafaxine XR 75 mg/day, or placebo for 8 weeks	MHT (Estrogen), anti-depressant (selective serotonin and norepine-phrine reuptake inhibitors (SSNRIs)	Treatment with low-dose venlafaxine and treatment with low-dose estradiol were each modestly more effective than placebo in reducing insomnia symptoms and improving subjective sleep quality.
Pinkerton, 2015 [59]	1174	Sleep disturbances—vasomotor symptoms	Paroxetine 7.5 mg or placebo for 24 weeks	SSRI	Paroxetine significantly reduces the number of nighttime awakenings attributed to vasomotor symptoms and increases sleep duration without differentially affecting sleep-onset latency or sedation.
Guthrie, 2018 [60]	546	Vasomotor symptoms (VMS), insomnia	Escitalopram 10–20 mg/day; yoga; aerobic exercise; 1.8 g/day omega-3 fatty acids; oral 17-beta-estradiol 0.5-mg/day; venlafaxine XR 75-mg/day; and cognitive behavioral therapy for insomnia (CBT-I).	Six different interventions: SSRI, exercise, MHT, serotonin-norepinephrine reuptake inhibitor—CBT-I	CBT-I as a first-line treatment in healthy midlife women with insomnia symptoms and moderately bothersome VMS.

FDA—Food and Drug Administration.

**Table 3 life-13-00649-t003:** Synopsis of the main studies of the neurologic symptoms.

**Authors,** **Year**	**Cases**	**Symptoms**	**Treatment**	**Category**	**Effect**
Anttalainen, 2013 [65]	34	Sleep disturbances, sleep-disordered breathing (SDB)	MPA (60 mg daily) or placebo for 14 days.	MHT (MPA)	In postmenopausal women with SDB, MPA induces a long-lasting stimulatory effect on breathing without improving sleep quality or the apnea-hypopnea index.
Kulkarni, 2018 [71]	44	Depressive symptoms	Tibolone (2.5 mg oral/day) or oral placebo for 12 weeks.	Selective tissue estrogenic activity regulator	Twelve weeks of tibolone significantly improved depression severity in women who presented with depression during menopause or postmenopause.
Moller, 2013 [68]	50	Cognitive fatigue	Estradiol valerate 2 mg + testosterone undecanoate 40 mg (E + T) daily or estradiol valerate 2 mg + placebo (E + P) daily.	MHT (Estrogen, Testosterone)	No significant treatment effect; E2/T ratio is important for optimal cognitive functioning.
Davari-Tanha, 2016 [66]	60	Sleep difficulty	Venlafaxine 75 mg/daily or citalopram 20 mg/daily or placebo Venlafaxine with the dose of 75 mg (began with 37.5 mg during the first week then increased to 75 mg) for 8 weeks; Citalopram with the dose of 20 mg (began with 10 mg during the first week and then increased to 20 mg) for 8 weeks; Placebo for 8 weeks.	Selective serotonin and norepinephrine reuptake inhibitors (SSNRIs)- venlafaxine and selective serotonin reuptake inhibitor (SSRI) class—citalopram	Citalopram and venlafaxine are equally more effective than placebo in reducing sleep disturbance and severity of hot flashes, while citalopram is more effective in reducing frequency of hot flashes than venlafaxine. Meanwhile, venlafaxine is more effective than citalopram in the treatment of depression in postmenopausal women.
Leeangkoon-sathian, 2017 [67]	100	Complains of insomnia	Group I—estradiol valerate 1 mg and dydrogesterone 10 mg. Group II—estradiol valerate 1mg and micronized progesterone 100 mg.	MHT (Estrogen, Progesterone)	Sleep quality improved in both groups; women in the micronized progesterone group had fewer overall side effects.
Gordon, 2018 [73]	172	Depressive symptoms	Transdermal estradiol (TE) (0.1 mg/day for 12 months) and intermittent micronized progesterone (IMP) (200 mg/d for 12 days).	MHT (Estrogen, Progesterone)	In perimenopausal and early postmenopausal women, TE + IMP administration prevents the increase in clinically transition-related depressive mood.
Sramek, 2017 [72]	209	Major depressive disorder	Lurasidone at 20–60 mg/day or placebo for 6 weeks.	Antipsychotic	Lurasidone was found to be effective in treating post-menopausal major depressive disorder.
Kornstein, 2014 [70]	426	Major depressive disorder	Desvenlafaxine 50 mg/day for 10 weeks vs. placebo.	SSRI	Desvenlafaxine 50 mg/day is effective in treating depression in both perimenopausal and postmenopausal women. Placebo response on measures of functional impairment is lower in perimenopausal women than in postmenopausal women, resulting in a greater apparent treatment benefit with desvenlafaxine among perimenopausal women.

**Table 4 life-13-00649-t004:** Synopsis of the main studies of musculoskeletal symptoms.

Authors, Year	Cases	Symptoms	Treatment	Category	Effect
Chlebowski, 2018 [74]	10,739	Joint symptoms	Daily oral conjugated equine estrogens (0.625 mg/d) or a matching placebo.	MHT (Estrogen)	Estrogen-alone use results in a modest but sustained reduction in the frequency of joint pain.
Cowan, 2022 [75]	132	Greater trochanteric pain syndrome	Transdermal creams containing oestradiol 50 mcg and norethisterone acetate 140 mcg.	MHT (Estrogen)	For postmenopausal women, especially with a BMI < 25, hormone therapy with any tendon-specific or sham exercise was superior to a placebo to reduce pain.

## Data Availability

Not applicable.

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
