# Peer review of "The Current Strategy in Hormonal and Non-Hormonal Therapies in Menopause—A Comprehensive Review"

_life, 2023, doi:10.3390/life13030649_

Round 1
Reviewer 1 Report
This manuscript presents a comprehensive literature review of the current clinical trials and practices in hormonal and non-hormonal therapies in menopause. It also provides insightful perspectives for future treatment strategies.
This reviewer recommends accepting this manuscript for publication after correcting the following typo errors.
1) Line 60 MHT: define this term.
2) Line 86 Material and Methods/data search: revise as Material and Methods (data search).
3) Line 266 SSRI: define this term.
4) Line 299 MPA: define this term.
Author Response
Dear Esteemed Reviewer,
Thank you for reviewing our paper.
- Line 60 MHT: define this term.
- Term defined in text as recommended - menopause hormonal therapy (MHT)
- Line 86 Material and Methods/data search: revise as Material and Methods (data search).
- Recommendation implemented
- Line 266 SSRI: define this term.
- Term defined in text as recommended -selective serotonine reuptake inhibitors (SSRI)
- Line 299 MPA: define this term.
- Term defined in text as recommended - medroxyprogesterone acetate [MPA]
Kindest regards,
From the behalf of authors,
Anca Pop

Reviewer 2 Report
The authors of this review performed great work regarding discussion of the most recent menopause studies on pharmacological resources, emerging therapies, and particularities of hormonal replacement therapy. I was pleased to read this paper. The paper is well written and all the references were used correctly.
Despite that I have some questions.
By which criteria have been evaluated the quality of the papers? And how did the authors used this information?
Also, the authors claim about using Microsoft Excel 2013 to perform statistical analysis but there is no any evidence of statistical analysis in the paper as it's a review. What have authors meant?
Author Response
Dear Esteemed Reviewer,
Thank you for reviewing our paper.
- By which criteria have been evaluated the quality of the papers? And how did the authors used this information?
- Thank you for your observation. The quality of the selected studies selected for review was evaluated based on study type randomized clinical trials (RCT) and clinical trials (CT). – change operated on line 117
- The authors claim about using Microsoft Excel 2013 to perform statistical analysis but there is no any evidence of statistical analysis in the paper as it's a review. What have authors meant?
- Thank you for your observation. We meant the data analysis process was accomplished by using Microsoft Excel program, Changed to The data analysis was performed using Microsoft Excel® 2013 - line
Kindest regards,
From the behalf of authors,
Anca Pop
